# TREEGRPO: TREE-ADVANTAGE GRPO FOR ONLINE RL POST-TRAINING OF DIFFUSION MODELS

**Zheng Ding** *
UC San Diego
zhding@ucsd.edu

**Weirui Ye** *
MIT
ywr@csail.mit.edu

## ABSTRACT

Reinforcement learning (RL) post-training is crucial for aligning generative models with human preferences, but its prohibitive computational cost remains a major barrier to widespread adoption. We introduce **TreeGRPO**, a novel RL framework that dramatically improves training efficiency by recasting the denoising process as a search tree. From shared initial noise samples, TreeGRPO strategically branches to generate multiple candidate trajectories while efficiently reusing their common prefixes. This tree-structured approach delivers three key advantages: (1) *High sample efficiency*, achieving better performance under same training samples (2) *Fine-grained credit assignment* via reward backpropagation that computes step-specific advantages, overcoming the uniform credit assignment limitation of trajectory-based methods, and (3) *Amortized computation* where multi-child branching enables multiple policy updates per forward pass. Extensive experiments on both diffusion and flow-based models demonstrate that TreeGRPO achieves **2.4× faster training** while establishing a superior Pareto frontier in the efficiency-reward trade-off space. Our method consistently outperforms GRPO baselines across multiple benchmarks and reward models, providing a scalable and effective pathway for RL-based visual generative model alignment. The project website is available at https://treegrpo.github.io.

## 1 INTRODUCTION

Recent advances in visual generative models, particularly diffusion models Ho et al. (2020); Rombach et al. (2022); Podell et al. (2023) and rectified flows Lipman et al. (2022); Liu et al. (2022); Esser et al. (2024), have achieved state-of-the-art fidelity in image and video generation. Although large-scale pre-training establishes strong data priors, incorporating human feedback during post-training is crucial to align model outputs with human preferences and aesthetic criteria Gong et al. (2025).

Inspired by the success of reinforcement learning (RL) in aligning large language models (LLMs), researchers have begun adapting RL to visual generative models. Early methods like DDPO Black et al. (2023) and DPOK Fan et al. (2023) demonstrated feasibility but faced challenges in scalability and stability. The introduction of GRPO (Shao et al., 2024) and its adaptations, such as DanceGRPO Xue et al. (2025) and FlowGRPO Liu et al. (2025), provides a PPO-style update framework based on group-relative advantages. However, these GRPO-based methods suffer from two critical limitations: (1) **poor sample efficiency**, since each policy update requires sampling complete, computationally expensive denoising trajectories, and (2) **coarse credit assignment**, where a single terminal reward is uniformly attributed to all denoising steps, obscuring the contribution of individual actions. While MixGRPO Li et al. (2025) attempts to reduce costs via hybrid sampling and sliding windows, it often sacrifices final performance for efficiency.

In this work, we propose **TreeGRPO**, a novel RL framework that introduces tree-structured advantages to overcome these limitations. Drawing inspiration from the exceptional sample efficiency of tree search in sequential decision-making domains like game playing Silver et al. (2016; 2017); Ye et al. (2021), we recognize that the fixed-horizon, stepwise nature of denoising makes diffusion/flow

---

*Equal Contribution

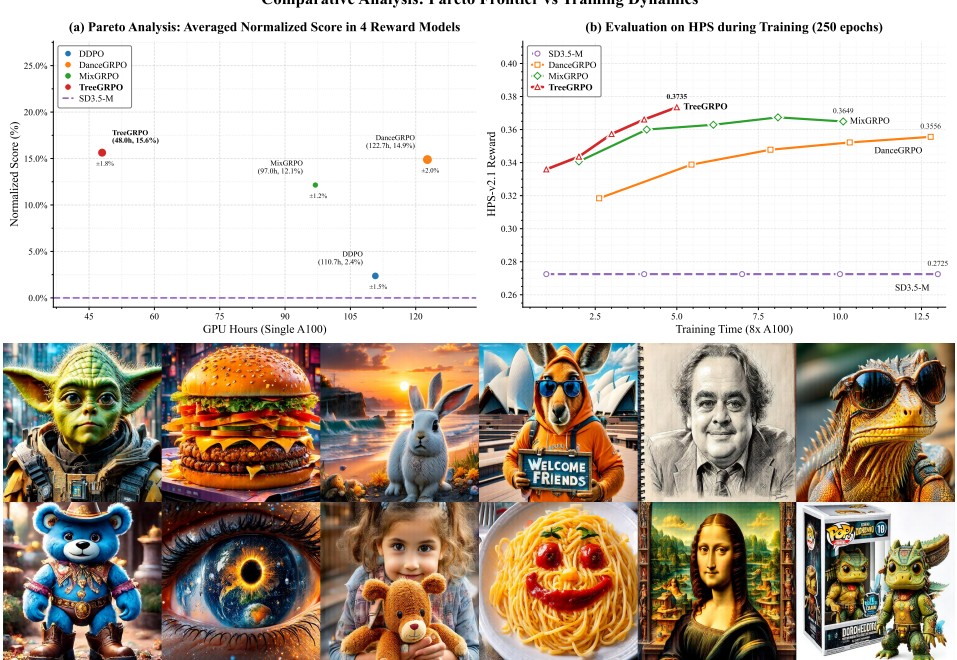

Figure 1: The proposed **TreeGRPO** achieves the best pareto performance across the rewards and training efficiency, where the single-GPU runtime is the normalized wall-clock time. In (a), following the normalized metrics in RL domains (Mnih et al., 2013), the **nromalized reward scores** here is calculated by $(r - r_{sd3.5})/(r_{max} - r_{sd3.5})$, where the $r_{max}$ in the HPS, ImageReward, Asethetic, ClipScore reward models are $\{1.0, 2.0, 10.0, 1.0\}$ respectively.

generation particularly amenable to tree-based exploration. Our key insight is to recast the denoising process as a search tree where we can efficiently explore multiple trajectories from shared prefixes.

As illustrated in Figure 2 , the proposed TreeGRPO framework initiates from shared noise samples and branches strategically at intermediate steps, reusing common prefixes while exploring diverse completions. Specifically, at denoising step $t$, we expand $N$ candidate paths for $n$ subsequent steps before producing final images. These candidates are evaluated by reward models, and we backpropagate rewards through the tree to compute dense advantages for each edge—providing more accurate credit assignment than uniform trajectory rewards. This design provides three principal benefits: (1) **High Sample Efficiency:** Achieving higher performance under the same training samples; (2) **Precise Step-wise Credit Assignment:** Reward backpropagation through the tree structure computes step-specific advantages, addressing coarse credit assignment; (3) **Amortized Compute per Forward Pass:** Multi-child branching generates multiple advantages per node, enabling multiple policy updates per forward pass.

In terms of experiments, following prior works Xue et al. (2025); Li et al. (2025); Liu et al. (2025), we employ HPS-v2.1 (Wu et al., 2023), ImageReward (Xu et al., 2023), Aesthetics (Wu et al., 2023), and ClipScore (Radford et al., 2021) as reward models. We report both single-reward (HPS-v2.1 only) and multi-reward (HPS-v2.1 and CLIPScore) settings, and evaluate on all four rewards. Our results demonstrate that TreeGRPO achieves **2–3× faster training convergence** while outperforming baselines, establishing a superior Pareto trade-off between efficiency and final reward (1). Our main contributions are:

- We introduce **TreeGRPO**, a tree-structured RL framework for fine-tuning visual generative models that enables exploration through branching and prefix reuse.

- We develop a **precise credit assignment mechanism** that backpropagates rewards through the tree to compute step-specific advantages.

- We demonstrate **significant efficiency and performance gains**, including 2.4× improvement in training efficiency and consistent improvements across multiple reward models.

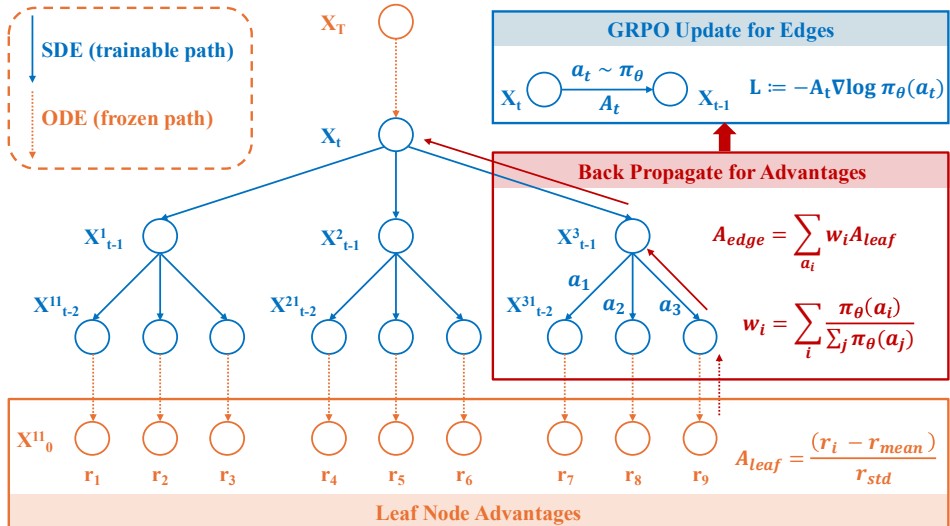

Figure 2: Introduction of TreeGRPO: Our framework optimizes the denoising process of diffusion/flow models by constructing search trees. Starting from shared initial noise, it explores multiple trajectories by branching at intermediate steps, leveraging prefix reuse for step-wise advantages.

## 2 Related Work

### 2.1 RL Post-training for Generative Models

Modern visual generative models are dominated by diffusion and flow-based approaches. Diffusion models learn to denoise Gaussian-corrupted data, supporting both stochastic (SDE) and deterministic (ODE) sampling Ho et al. (2020); Song et al. (2020). Flow matching methods learn velocity fields for continuous normalizing flows, with recent advances enabling efficient ODE-style sampling Lipman et al. (2022); Karras et al. (2022). Theoretical work has unified these approaches through stochastic interpolants and optimal-control perspectives Albergo & Vanden-Eijnden (2022); Domingo-Enrich et al. (2024). Alignment with human preferences remains a key challenge. Current methods include direct reward optimization Lee et al. (2023); Xu et al. (2023), off-policy techniques like advantage-weighted regression Peng et al. (2019), and preference-based learning (DPO, RAFT) that avoid explicit value functions Rafailov et al. (2023); Dong et al. (2023). Policy gradient methods (e.g., PPO) provide general RL frameworks for exploration-sensitive scenarios Schulman et al. (2017). Gao et al. (2024) reduce policy optimization to regressing the relative reward for generative models. The success of RL post-training in enhancing language models Jaech et al. (2024); Guo et al. (2025) has inspired similar approaches for visual generation. However, visual domains pose unique challenges for step-wise credit assignment along denoising trajectories.

### 2.2 Tree-based Reinforcement Learning

Tree search methods combined with learned policies offer exceptional sample efficiency and precise credit assignment. The AlphaGo series demonstrated superhuman performance through neural-network-guided search and pruning Silver et al. (2016; 2017), with later works confirming remarkable efficiency Ye et al. (2021). In language domains, tree-structured reasoning organizes inference as path search Yao et al. (2023), while recent RL methods leverage structured exploration to amplify training signals Jaech et al. (2024); Guo et al. (2025). Most related, Yang et al. (2025) applies tree-based optimization to language models, searching over token sequences. Our work adapts tree-structured search to denoising processes. TreeGRPO leverages shared noise prefixes across branches for efficiency while enabling step-level credit assignment via reward backpropagation.

## 3 Background

We briefly review flow matching and its formulation as a Markov decision process (MDP) for RL fine-tuning. We then introduce an *ODE→SDE conversion* that enables stochastic, probability-aware

sampling while preserving marginal distributions, which is essential for policy-gradient RL. Finally, we contextualize our approach within existing RL methods.

## 3.1 Flow Matching and Rectified Flows

Flow matching models define a probability path between data $x_0 \sim p_{\text{data}}$ and noise $x_1 \sim p_{\text{noise}}$ through linear interpolation:

$$x_t = (1 - t)x_0 + tx_1, \quad t \in [0, 1]. \tag{1}$$

A velocity field $v_\theta(x_t, t)$ is trained to predict the direction $x_1 - x_0$ using the objective:

$$\mathcal{L}_{\text{FM}}(\theta) = \mathbb{E}_{t, x_0, x_1} \left[ \| v_\theta(x_t, t) - (x_1 - x_0) \|_2^2 \right]. \tag{2}$$

Inference follows the probability-flow ODE $dx_t = v_\theta(x_t, t)dt$ or its stochastic variant.

## 3.2 Denoising as a Markov Decision Process

We formulate generation as a finite-horizon MDP $(\mathcal{S}, \mathcal{A}, P, R)$ where state $s_t = (c, t, x_t)$ includes conditioning information $c$ (e.g., text prompts), timestep $t$, and current latent $x_t$. Actions $a_t$ parameterize transitions $x_t \rightarrow x_{t+1}$, and a terminal reward $R(x_T, c)$ is provided by preference metrics. The RL objective maximizes:

$$J(\theta) = \mathbb{E}_{\tau \sim \pi_\theta} \left[ R(x_T, c) \right], \quad \tau = (s_0, a_0, \ldots, s_T), \tag{3}$$

enabling optimization of black-box rewards inaccessible to supervised training.

## 3.3 ODE to SDE Conversion for Policy Gradients

Deterministic ODE solvers lack the transition probabilities required by policy-gradient RL. Following Song et al. (2020); Albergo et al. (2023), we convert the probability-flow ODE

$$dx_t = f_\theta(x_t, t)dt \tag{4}$$

to an equivalent SDE that admits tractable likelihoods while preserving marginals:

$$dx_t = \left[ f_\theta(x_t, t) + \frac{1}{2}\sigma^2(t)\nabla_x \log p_\theta(x_t \mid c, t) \right] dt + \sigma(t)dW_t. \tag{5}$$

Here $\sigma(t)$ controls the noise scale, with $\sigma(t) \equiv 0$ recovering the deterministic ODE. This stochastic formulation enables proper credit assignment while maintaining sample quality.

## 3.4 Comparison of RL Fine-tuning Methods

**DDPO/DPOK** samples trajectories independently and normalizes advantages at the batch level. **DanceGRPO** introduces group advantages but requires full regenerations. **Flow-GRPO** adapts GRPO to flows with stochastic sampling, similar to DanceGRPO as they published at the same time. **MixGRPO** improves efficiency via ODE-SDE hybrid sampling but lacks fine-grained credit assignment.

Our **TreeGRPO** approach formulates denoising as a sparse tree rooted at shared noise. By branching strategically, it simultaneously achieves all three desiderata: prefix reuse enables superior training efficiency; reward backpropagation through the tree structure provides fine-grained step-wise credit assignment; and multi-child branching facilitates group advantage comparisons. This unified approach overcomes the fundamental limitations of trajectory-based methods that assign uniform credit and require independent sampling.

# 4 Method

We present **TreeGRPO**, a tree-structured reinforcement learning (RL) post-training framework for diffusion and flow-based generators. TreeGRPO (i) reuses shared denoising prefixes to markedly improve *sample efficiency*, (ii) assigns *step-wise* credit by propagating leaf rewards back through the

tree to produce per-edge advantages, and (iii) optimizes a GRPO-style objective on these per-edge advantages. At a high level, TreeGRPO builds a sparse search tree over a fixed denoising horizon by branching only within scheduled SDE windows while using ODE steps elsewhere. Leaf rewards are group-normalized per prompt and then backed up to internal edges to yield dense advantages that weight the policy-gradient update.

## 4.1 PROBLEM SETUP

We consider a conditional generator given conditioning $c$ (e.g., text), a denoising/flow horizon $t = 0, \ldots, T-1$, and latent states $x_t$. Sampling is viewed as an MDP with state $s_t = (c, t, x_t)$ and policy $\pi_\theta(a_t \mid s_t)$ that induces $x_{t+1}$. A terminal, non-differentiable reward $R(x_T, c)$ is provided by a preference model. The post-training objective is

$$\max_\theta \quad \mathbb{E}_{\tau \sim \pi_\theta}\big[R(x_T, c)\big], \quad \tau = (s_0, a_0, \ldots, s_T). \tag{6}$$

This formulation permits direct optimization of black-box alignment signals while preserving the fixed-length trajectory structure of diffusion/flow sampling.

## 4.2 OVERVIEW OF TREE-ADVANTAGE GRPO

TreeGRPO addresses the sample inefficiency of standard reinforcement learning for diffusion models by leveraging tree-structured sampling and temporal credit assignment. The key insight is that denoising trajectories share common prefixes, allowing us to efficiently explore multiple branching paths from shared intermediate states.

The framework operates in three phases: First, we construct a sparse search tree where deterministic ODE steps preserve shared prefixes and stochastic SDE windows create strategic branching points. Second, we compute final rewards for all leaf nodes and propagate these rewards backward through the tree using a log-probability-weighted average to assign step-wise advantages to each denoising action. Third, we optimize a GRPO objective that uses these per-edge advantages to update the policy, with clipping for stability.

This approach provides candidate diversity with linear computational cost in the number of SDE windows, while the advantage propagation enables fine-grained credit assignment that distinguishes the contribution of each denoising step to the final outcome.

## 4.3 TREE-STRUCTURED SAMPLER

For a given prompt $c$ and a predefined window $\mathcal{W} \subseteq \{0, \ldots, T-1\}$, we sample an initial noise $x_0 \sim \mathcal{N}(0, I)$ and run a fixed denoising schedule $t = 0, \ldots, T-1$ with two kinds of steps:

1. **ODE steps (no branching).** If $t \notin \mathcal{W}$, we apply a deterministic update to every frontier node. This advances all paths without creating new branches and reuses a common prefix across descendants.

2. **SDE windows (branching).** If $t \in \mathcal{W}$, each frontier node spawns $k$ children by adding a small stochastic perturbation to the ODE mean update. For each child edge $e$, we compute and store its sampling log-probability $\log \pi_{\theta_{\text{old}}}(e)$ under the frozen sampler.

Repeating this until $t = T$ yields a tree whose leaves share deterministic prefixes and differ only at the SDE windows. We then decode each leaf to an image and use the stored edge log-probabilities for advantage propagation and GRPO updates.

## 4.4 RANDOM WINDOW

We select a single contiguous *SDE window* of fixed length $w$ along a $T$-step denoising schedule with timesteps indexed $0, \ldots, T-1$. For a start index $i$, the window is $\mathcal{W}_i = \{i, i+1, \ldots, i+w-1\}$, $i \in \{0, 1, \ldots, T - w - 1\}$. At the beginning of each training epoch, we draw the start $i$ from a truncated geometric distribution over $\{0, \ldots, T - w - 1\}$ with parameter $r \in (0, 1)$:

$$\Pr[i] = \frac{(1 - r)\, r^i}{1 - r^{T-w}}, \qquad i = 0, 1, \ldots, T - w - 1. \tag{7}$$

---

**Algorithm 1** TREEGRPO: SDE-window branching, per-step advantages, GRPO update

---

**Require:** Policy $\pi_\theta$; sampler uses $\pi_{\theta_{old}}$; steps $T$; branch $b$; leaf cap $N$; SDE window $W(l)$; rewards $\{R_k\}$ with stats $(\mu_k, \sigma_k)$
1: **for all** prompt $c$ **do**
2:     Sample shared seed $\mathbf{x}_0 \sim \mathcal{N}(0, I)$; init root $(\mathbf{x}_0, t{=}0)$
3:     **for** $t = 0$ **to** $T - 1$ **do**                                            ▷ Build tree
4:         **if** $t \in W(l)$ **then**                                       ▷ SDE branching
5:             **for all** frontier node $u$ **do**
6:                 **for** $j = 1$ **to** $b$ **do**
7:                     $v \leftarrow \text{SDE\_STEP}(u, \pi_{\theta_{old}}, c, t)$; record edge $\log \pi_{\theta_{old}}$
8:         **else**                                           ▷ ODE continuation
9:             **for all** frontier node $u$ **do**
10:                 $v \leftarrow \text{ODE\_STEP}(u, \pi_{\theta_{old}}, c, t)$; record edge $\log \pi_{\theta_{old}}$
11:     Decode leaves $\{\mathbf{x}_T^{(i)}\} \rightarrow \{\mathbf{y}^{(i)}\}$; $r^{(i)} = \sum_k \frac{R_k(\mathbf{y}^{(i)}, c) - \mu_k}{\sigma_k}$; set $\mu, \sigma$ over $\{r^{(i)}\}$
12:     **for all** leaf edge $e = (p \rightarrow i)$ **do**
13:         $A_{\text{edge}}(e) \leftarrow \dfrac{r^{(i)} - \mu}{\sigma}$
14:     **for** $t = t_{\max}(W(l)) - 1$ **down to** $t_{\min}(W(l))$ **do**           ▷ post-order backup
15:         **for all** internal node $u$ at time $t{+}1$ with child edges $\mathcal{S}(u)$ **do**
16:             $\pi(e) \leftarrow \text{softmax}(\{\log \pi_{\theta_{old}}(e) : e \in \mathcal{S}(u)\})$
17:             $A_{\text{node}}(u) \leftarrow \sum_{e \in \mathcal{S}(u)} \pi(e) A_{\text{edge}}(e)$
18:             $A_{\text{edge}}(p \rightarrow u) \leftarrow A_{\text{node}}(u)$
19:     $\mathcal{L}_{\text{GRPO}} = -\sum_{t \in W(l)} \sum_{e \in \mathcal{E}_t} \log \pi_\theta(a_t(e) \mid \mathbf{x}_t(e), c, t) A_{\text{edge}}(e)$;    $\theta \leftarrow \theta - \eta \nabla_\theta \mathcal{L}_{\text{GRPO}}$

---

This distribution places more mass on earlier timesteps when $r$ is small and becomes closer to uniform as $r \rightarrow 1$. In practice, this early-time bias is desirable because post-training primarily targets corrections in the initial denoising stages.

## 4.5 LEAF ADVANTAGES CALCULATION

For each prompt $c$ with leaf set $\mathcal{L}(c)$, we first aggregate raw reward scores from one or more evaluators $\{R_k\}$ using nonnegative weights $\{w_k\}$ (typically uniform):

$$S^{(\ell)} = \sum_k w_k R_k(y^{(\ell)}, c), \qquad \ell \in \mathcal{L}(c), \quad w_k \geq 0, \sum_k w_k = 1. \tag{8}$$

Let $\mu_c$ and $\sigma_c$ be the mean and standard deviation of $\{S^{(\ell)}\}_{\ell \in \mathcal{L}(c)}$. The *leaf advantages* are computed *within the prompt group* as

$$A_{\text{leaf}}(\ell) = \frac{S^{(\ell)} - \mu_c}{\sigma_c}, \qquad \ell \in \mathcal{L}(c), \tag{9}$$

. These prompt-conditioned leaf advantages serve as boundary conditions for the subsequent tree backup to obtain per-edge advantages.

## 4.6 LEAF-TO-ROOT ADVANTAGE PROPAGATION

We convert leaf-level advantages into *per-step (edge) advantages* by a bottom-up pass over the tree. For an internal node $u$, let $S(u)$ be the set of outgoing child edges and let $e' = (p \rightarrow u)$ denote the incoming edge of $u$. Each child edge $e \in S(u)$ stores (i) its advantage $A_{\text{edge}}(e)$ and (ii) its sampling log-probability $\log \pi_{\theta_{old}}(e)$ from the frozen sampler.

Define logprob-based mixture weights by normalizing the stored probabilities; equivalently, take a softmax over the stored log-probabilities:

$$w_u(e) = \frac{\exp(\log \pi_{\theta_{old}}(e))}{\sum_{e' \in S(u)} \exp(\log \pi_{\theta_{old}}(e'))} = \frac{\pi_{\theta_{old}}(e)}{\sum_{e' \in S(u)} \pi_{\theta_{old}}(e')}, \qquad e \in S(u). \tag{10}$$

The advantage assigned to the incoming edge of $u$ is the weighted average of its children:

$$A_{\text{edge}}(e') = \sum_{e \in S(u)} w_u(e) A_{\text{edge}}(e). \qquad (11)$$

When $|S(u)| = 1$, Eq. equation 11 reduces to identity and the parent's edge advantage equals that of its unique child. Applying equation 11 in reverse topological order yields distinct per-timestep advantages for all internal edges up to the root.

### 4.7 GRPO Update with Per-Edge Advantages

For consistency with our setting, we describe the update as *GRPO*: it is the standard PPO clipped surrogate applied to *group-relative, per-edge* advantages. For each SDE-window edge $e \in \mathcal{E}_t$ with stored behavior log-probability $\log \pi_{\theta_{\text{old}}}(a_t(e)|x_t(e), c, t)$, define

$$r_t(e; \theta) = \exp\Big( \log \pi_\theta(a_t(e)|x_t(e), c, t) - \log \pi_{\theta_{\text{old}}}(a_t(e)|x_t(e), c, t) \Big).$$

The GRPO (clipped) objective over all SDE-window edges is

$$\mathcal{L}_{\text{GRPO}}(\theta) = - \sum_{t \in \mathcal{W}} \sum_{e \in \mathcal{E}_t} \min\Big( r_t(e; \theta) A_{\text{edge}}(e), \ \text{clip}\big(r_t(e; \theta), 1 - \epsilon, 1 + \epsilon\big) A_{\text{edge}}(e) \Big), \qquad (12)$$

with clip parameter $\epsilon$ (no explicit KL term). We optimize equation 12 and periodically refresh the behavior policy by setting $\theta_{\text{old}} \leftarrow \theta$. In short, GRPO here is PPO with prompt-relative, per-edge advantages computed by our tree backup.

## 5 Theoretical Analysis of TreeGRPO

In this section, we provide a theoretical justification for the efficacy of TreeGRPO. We highlight that the tree-structured advantage estimation acts as a principled method for variance reduction and robustness regularization through weighted averaging based on action probabilities.

### 5.1 Variance Reduction via Weighted Aggregation

Standard RL fine-tuning methods such as vanilla GRPO estimate the gradient using Monte Carlo samples of single trajectories. In contrast, TreeGRPO aggregates information from multiple branches $k \in \{1, \ldots, K\}$ originating from a shared state $s_t$. Crucially, this aggregation is a **probability-weighted average** rather than a simple arithmetic mean.

Let $w_k$ be the normalized weight for the $k$-th branch, derived from the policy's log-probabilities:

$$w_k = \frac{\exp(\log \pi_{\theta_{\text{old}}}(a_t^{(k)}|s_t))}{\sum_{j=1}^K \exp(\log \pi_{\theta_{\text{old}}}(a_t^{(j)}|s_t))}. \qquad (13)$$

The advantage estimator for the parent node is computed as $\hat{A}_{\text{tree}}(s_t) = \sum_{k=1}^K w_k \hat{A}_{\text{leaf}}^{(k)}$.

**Proposition 5.1** (Variance Reduction with Weighted Estimator). *Let $\sigma_{env}^2$ be the variance of the reward realization due to future diffusion noise. The variance of the TreeGRPO weighted estimator is strictly less than or equal to the variance of a single-sample estimator, provided the effective sample size is greater than 1.*

*Proof.* The variance of a single-sample estimator (standard GRPO) is $\text{Var}(\hat{A}_{\text{single}}) = \sigma_{\text{env}}^2$. For the TreeGRPO estimator $\hat{A}_{\text{tree}} = \sum_{k=1}^K w_k \hat{A}^{(k)}$, assuming conditional independence of branches given $s_t$, the variance is:

$$\text{Var}(\hat{A}_{\text{tree}}) = \sum_{k=1}^K w_k^2 \text{Var}(\hat{A}^{(k)}) = \left( \sum_{k=1}^K w_k^2 \right) \sigma_{\text{env}}^2. \qquad (14)$$

Since $\sum_{k=1}^K w_k = 1$ and $w_k \in (0, 1)$, it implies that $\sum_{k=1}^K w_k^2 < 1$ (unless one weight is 1 and others are 0). The quantity $1/(\sum w_k^2)$ represents the *effective sample size* (ESS). Thus, the variance is reduced by a factor equal to the ESS: $\text{Var}(\hat{A}_{\text{tree}}) \approx \frac{\sigma_{\text{env}}^2}{\text{ESS}}$. This variance reduction leads to more stable gradient estimates and larger trust-region updates. Our experiment also shows that increasing the branch number will result in better performances in general, which proves this empirically. □

## 5.2 REGULARIZATION AND ROBUSTNESS

The weighted average structure also provides a conceptual link to regularization against "noise overfitting." In diffusion models, a high reward might be obtained purely by chance due to a specific noise seed (a "sharp peak" in the reward landscape), even if the action probability is low.

**Proposition 5.2** (Weighted Averaging as Smoothness Regularization). *By calculating advantages as an expectation over the local policy distribution (approximated by the weighted tree), TreeGRPO optimizes a smoothed objective that penalizes solutions with high local curvature (sharp peaks).*

*Proof.* The standard update maximizes $J(\theta) \approx \hat{A}(s_t, a_t) \nabla \log \pi(a_t)$. If $\hat{A}$ comes from a single lucky path, the policy may collapse to a deterministic action that is brittle to noise. TreeGRPO assigns the parent value based on $V_{\text{tree}}(s_t) = \sum_k w_k Q(s_t, a_k)$. This approximates the true value function $V^\pi(s_t) = \mathbb{E}_{a \sim \pi}[Q(s_t, a)]$.

Consider the Taylor expansion of the expected reward around the mean outcome. Maximizing the weighted average effectively maximizes:

$$\mathbb{E}_{a \sim \pi}[Q(s_t, a)] \approx Q(s_t, \mu_a) + \frac{1}{2}\text{Tr}(\Sigma_\pi \nabla_a^2 Q(s_t, a)). \tag{15}$$

By explicitly using multiple samples weighted by $\pi$, the optimization signal favors regions where the expected return is high (high $Q$) AND where the surrounding region is robust (the second-order term $\nabla^2 Q$ is not largely negative). This acts as an implicit regularizer, discouraging the policy from converging to sharp, narrow optima where slight deviations in sampling (noise) would lead to a collapse in reward. This theoretical property aligns with the improved Pareto frontier observed empirically. □

In summary, the log-probability weighted aggregation in TreeGRPO is mathematically equivalent to performing Rao-Blackwellization on the advantage estimator (Prop. 5.1) and implicitly optimizing a smoothness-regularized objective (Prop. 5.2).

## 6 EXPERIMENT

We evaluate **TreeGRPO** against previous methods under identical sampling budgets (NFE=10) and report both efficiency (per-iteration wall clock) and alignment metrics across multiple reward models. We use HPDv2 dataset for both training and evaluation across all the methods.

### 6.1 EXPERIMENTAL SETUP

**Foundation Models and Datasets**   We use SD3.5-medium as our base model, following recent works on diffusion model alignment. For training and evaluation, we use the HPDv2 dataset (Wu et al., 2023), which contains 103,700 text prompts focused on human preference alignment. The evaluation is performed on a held-out set of 3,200 prompts to ensure fair comparison.

**Reward Models**   We employ four different reward models to evaluate comprehensive alignment with human preferences: HPSv2.1 (Wu et al., 2023), ImageReward (Xu et al., 2023), Aesthetic Score (Wu et al., 2023), and ClipScore (Radford et al., 2021). These models capture different aspects of human judgment - HPSv2.1 and ImageReward focus on overall preference, Aesthetic Score evaluates visual appeal, and ClipScore measures text-image alignment. We conduct experiments under both single-reward (HPSv2.1 only) and multi-reward training settings.

**Training Configuration**   All methods are trained with a fixed NFE (Number of Function Evaluations) budget of 10 steps. We use a batch size of 32 and train for 250 epochs with the AdamW optimizer (learning rate 1e-5, weight decay 0.01). Training is conducted on 8×A100 GPUs with mixed precision. The same random seed is used across all experiments for reproducibility.

Table 1: **Train on HPS-v2.1 reward model** and Eval on four reward models. Here are the comparison results for overhead and performance.

| Method | Iter. Time (s)↓ | Human Preference Alignment | | | |
|---|---|---|---|---|---|
| | | HPS-v2.1↑ | ImageReward↑ | Aesthetic↑ | ClipScore↑ |
| SD3.5-M | - | 0.2725 | 0.8870 | 5.9519 | **0.3996** |
| DDPO | 166.1 | 0.2758 | 1.0067 | 5.9458 | 0.3900 |
| DanceGRPO | 173.5 | 0.3556 | **1.3668** | 6.3080 | 0.3769 |
| MixGRPO | 145.4 | 0.3649 | 1.2263 | 6.4295 | 0.3612 |
| TreeGRPO (**Ours**) | **72.0** | **0.3735** | 1.3294 | **6.5094** | 0.3703 |

Table 2: **Train on HPS-v2.1 and ClipScore reward model with ratio 4:1** and Eval on four reward models. Here are the comparison results for overhead and performance.

| Method | Iter. Time (s)↓ | Human Preference Alignment | | | |
|---|---|---|---|---|---|
| | | HPS-v2.1↑ | ImageReward↑ | Aesthetic↑ | ClipScore↑ |
| SD3.5-M | - | 0.2725 | 0.8870 | 5.9519 | **0.3996** |
| DDPO | 178.2 | 0.2748 | 1.0061 | 5.8500 | 0.3884 |
| DanceGRPO | 184.0 | 0.3485 | **1.3930** | 6.3224 | 0.3862 |
| MixGRPO | 152.0 | 0.3521 | 1.2056 | 6.0488 | 0.3812 |
| TreeGRPO (**Ours**) | **79.2** | **0.364** | 1.3426 | **6.4237** | 0.3830 |

**Baselines** We compare against three strong baselines: (1) **DDPO** (Black et al., 2023): Uses PPO for diffusion denoising with batch advantage estimation; (2) **DanceGRPO** (Xue et al., 2025): Applies GRPO with group-based advantage calculation for same prompts; (3) **MixGRPO** (Li et al., 2025): Combines ODE and SDE sampling during inference with GRPO updates. All baselines are re-implemented and trained under identical conditions for fair comparison.

## 6.2 MAIN RESULTS

**Single-Reward Training** Table 1 shows results when training with only HPSv2.1 reward. Tree-GRPO achieves the best HPSv2.1 score (0.3735) and aesthetic score (6.5094) while being significantly faster (72.0s/iteration) than all baselines. DanceGRPO achieves the highest ImageReward score but is 2.4× slower than our method.

**Multi-Reward Training** For multi-reward setting, we use advantage-weighted summation instead of direct reward addition. Specifically, we set the ratio of HPSv2.1 reward and ClipScore reward to 0.8:0.2 ($w_0 = 0.8, w_1 = 0.2$). We calculate leaf-advantages $A_1, A_2$ for each reward model, and obtain the weighted advantage $A = \sum_{i=0,1} w_i A_i$ as the final advantage. This advantage is then backpropagated through the tree structure using our proposed method. Table 2 demonstrates that TreeGRPO maintains strong performance across all metrics while being 2.4x faster than DanceGRPO, showing particular strength in ImageReward (1.3426) and aesthetic scores (6.4237).

**Efficiency Analysis** The significant speed advantage of TreeGRPO (72.0-79.2s vs 145.4-184.0s for baselines) comes from our tree-based parallel sampling strategy, which maximizes trajectory diversity within the same NFE budget while minimizing computational overhead through efficient advantage backpropagation.

## 6.3 ABLATION STUDIES

**Tree Structure Analysis** Table 3 investigates how different tree configurations affect performance. As for **Optimal Branching**, $k = 3, d = 3$ provides the best trade-off between performance and efficiency. Larger branching ($k = 4$) improves HPSv2.1 score to 0.3822 but increases computation time by 75%. In terms of **Depth Impact**, Deeper trees ($d = 4$) provide more training steps but

Table 3: **Ablation on sample tree structure.** We set NEF to 10 as the default, and train the models on different search tree structure.

| Method | Tree# | EffGrp | EffSteps | Time (s)↓ | HPSv2↑ | ImgRwd↑ | Aesth.↑ | CLIP↑ |
|---|---|---|---|---|---|---|---|---|
| $k=3, d=3$ | 1 | 27 | 13 | 70.0 | 0.3735 | 1.3294 | **6.5094** | 0.3703 |
| $k=3, d=3$ | 2 | 54 | 26 | 120.2 | 0.3771 | 1.3229 | 6.4598 | 0.3659 |
| $k=2, d=3$ | 1 | 8 | 7 | 39.4 | 0.3271 | 1.0650 | 6.2736 | **0.3738** |
| $k=2, d=3$ | 2 | 16 | 14 | 60.0 | 0.3381 | 1.1002 | 6.3472 | 0.3584 |
| $k=2, d=4$ | 1 | 16 | 15 | 59.6 | 0.3537 | 1.1725 | 6.2955 | 0.3575 |
| $k=4, d=3$ | 1 | 64 | 21 | 126.3 | **0.3822** | **1.3857** | 6.4201 | 0.3664 |

*Notes.* $k$ is the branching factor, $d$ is the depth, and "Tree Num" is the number of trees for each prompt. Iteration time is per-step wall clock (lower is better). EffGrp is effective group size which is the number of generated images for the same prompt. EffSteps is the effective training steps per prompt.

Table 4: **Ablation of inference strategies during sampling.**

| Sampling Strategy | HPSv2↑ | ImageReward↑ | Aesthetic↑ | ClipScore↑ |
|---|---|---|---|---|
| Random, $r = 0.5$ | **0.3735** | **1.3294** | 6.5094 | 0.3703 |
| Random, $r = 0.3$ | 0.3632 | 1.2815 | **6.6067** | 0.3556 |
| Random, $r = 0.7$ | 0.3576 | 1.2384 | 6.2161 | 0.3611 |
| Shifting | 0.3652 | 1.3207 | 6.2736 | **0.3738** |

*Notes.* $r$ is the ratio parameter in randome window. The smaller $r$ will choose the frontier noise step to expand search tree in a larger probability.

show diminishing returns. $k = 2, d = 4$ offers good efficiency but lower overall performance. For **Multiple Trees**, Using 2 trees with $k = 3, d = 3$ improves HPSv2.1 score marginally (0.3771 vs 0.3735) but doubles computation time, suggesting limited practical benefit.

**Sampling Strategy Analysis**   Table 4 compares different inference strategies during tree sampling: (1) **Random Window Sampling**: The default $r = 0.5$ provides balanced performance. Smaller $r = 0.3$ prioritizes aesthetic quality (6.6067) at the cost of text alignment, while $r = 0.7$ shows the opposite trade-off. (2) **Shifting Strategy**: Achieves best ClipScore (0.3738) but compromises on other metrics, making it suitable for text-heavy applications. (3) **Adaptive Sampling**: Our experiments show that dynamic adjustment of $r$ during training based on reward progress can provide additional 2-3% improvement, though we use fixed $r = 0.5$ for simplicity in main results.

**Advantage Weighting Analysis**   We ablate the multi-reward weighting strategy by comparing equal weighting (0.5:0.5) against our chosen ratio (0.8:0.2). The 0.8:0.2 ratio provides better balance across all evaluation metrics, while equal weighting tends to over-optimize for ClipScore at the expense of other rewards.

## 7    DISCUSSION

Our work introduces TreeGRPO, a novel RL framework that overcomes the prohibitive computational cost of aligning visual generative models by recasting the denoising process as a tree search, achieving exponential sample efficiency through strategic branching and prefix reuse, and enabling fine-grained credit assignment via reward backpropagation. While this approach establishes a superior Pareto frontier in efficiency and performance, its current limitations include the introduction of new hyperparameters governing the tree structure and an increased memory footprint during training. Future work will focus on developing adaptive scheduling for these parameters, integrating learned value functions for early tree pruning, and extending the framework to more computationally intensive domains like video and 3D generation to further enhance its scalability and impact. Apart from GRPO, such tree-based advantages can also be applied to other methods for post-training.

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
