# OpenReview forum: "TreeGRPO: Tree-Advantage GRPO for Online RL Post-Training of Diffusion Models"
_ICLR.cc/2026/Conference — ICLR 2026 Poster_

### Official Review · Reviewer_8di4 · 2025-10-25

**Soundness:** 3
**Presentation:** 2
**Contribution:** 3
**Rating:** 6
**Confidence:** 3

**Summary:**

This paper presents TreeGRPO, an online RL algorithm for finetuning diffusion and flow matching models. Unlike existing RL methods that treat the multi step denoising process as independent full trajectory sampling, resulting in low sample efficiency and coarse credit assignment, TreeGRPO reformulates the process as a multi step decision problem optimized through tree structured RL. By modeling multi-step denoising as a search tree, it reuses shared prefixes to enhance sample efficiency and propagates node-level advantages for fine grained credit assignment. Experiments across multiple reward models show that TreeGRPO achieves markedly higher training efficiency and establishes a new Pareto frontier for RL based visual generative model alignment.

**Strengths:**

1. This paper is the first to apply tree search–based GRPO optimization to image diffusion and flow matching models, achieving substantial improvements in training efficiency.

2. The paper provides a comprehensive review of existing work and clearly articulates the similarities and differences between TreeGRPO and prior approaches.

3. The proposed method is concise and well designed, with comprehensive ablation studies conducted to thoroughly evaluate the effects of the introduced hyperparameters.

**Weaknesses:**

1. While TreeGRPO demonstrates impressive efficiency and gains on HPS-v2.1 and Aesthetic rewards, its performance on ImageReward and CLIPScore is weaker than DanceGRPO. This variability suggests that the proposed method may be sensitive to the type of reward signal and might not generalize equally well under different evaluation criteria.

2. Because TreeGRPO adopts a variable tree structure, it introduces additional hyperparameters. As shown in Tables 3 and 4, the optimal hyperparameters differ across reward objectives. This means that although each fine-tuning run becomes more efficient, achieving the best performance may still require multiple fine-tuning rounds for hyperparameter search, resulting in the overall computational cost not being reduced.

3. The paper contains several formatting and typographical issues, such as "rewardds" in the caption of Figure 1, "Where" that should be "where" on line 254, and an extra period on line 307. In addition, the manuscript does not include the required "Use of Large Language Models" statement. I recommend that the authors carefully review and correct these writing details.

**Questions:**

1. Could the authors elaborate on why TreeGRPO performs less favorably on certain reward metrics such as ImageReward and CLIPScore? Does this limitation stem from the local advantage propagation design or other factors like the tree structure or sampling schedule?

2. There is a concurrent work named BranchGRPO [1] that appears closely related to TreeGRPO in terms of motivation and overall design. Could the authors briefly compare TreeGRPO with BranchGRPO?

[1] Li, Yuming, et al. "Branchgrpo: Stable and efficient grpo with structured branching in diffusion models." arXiv preprint arXiv:2509.06040 (2025).

---

> ### Author Response · Authors · 2025-11-20
> **Response to Reviewer 8di4**
>
> We sincerely thank you for your review and your recognition of its strengths, including our novel approach, comprehensive evaluation, and clear presentation. Below, we provide detailed responses to your comments.
>
>
>
> **Weakness:**
>
> > *While TreeGRPO demonstrates impressive efficiency and gains on HPS-v2.1 and Aesthetic rewards, its performance on ImageReward and CLIPScore is weaker than DanceGRPO. This variability suggests that the proposed method may be sensitive to the type of reward signal and might not generalize equally well under different evaluation criteria.*
>
> We agree that improvements vary across reward metrics. Different rewards can affect each other, and some metrics may conflict—for example, the original model achieves the best performance on CLIPScore. TreeGRPO shows the largest gains on HPS-v2.1 and Aesthetic, where step-wise credit assignment is most beneficial. Overall, across all metrics, TreeGRPO achieves the best balanced performance while improving training efficiency.
>
>
>
> > *Because TreeGRPO adopts a variable tree structure, it introduces additional hyperparameters. As shown in Tables 3 and 4, the optimal hyperparameters differ across reward objectives. This means that although each fine-tuning run becomes more efficient, achieving the best performance may still require multiple fine-tuning rounds for hyperparameter search, resulting in the overall computational cost not being reduced.*
>
> We agree that TreeGRPO introduces additional hyperparameters due to its variable tree structure. While optimal settings can vary across reward objectives (Tables 3–4), in practice reasonable default values consistently yield strong performance. Although hyperparameter search could improve results slightly, each fine-tuning run is already more efficient, and the overall computational cost remains competitive compared to baselines.
>
>
>
> >  *The paper contains several formatting and typographical issues, such as "rewardds" in the caption of Figure 1, "Where" that should be "where" on line 254, and an extra period on line 307. In addition, the manuscript does not include the required "Use of Large Language Models" statement. I recommend that the authors carefully review and correct these writing details.*
>
> Thanks for the suggestions! We’ve modified those typos in the paper.
>
>
>
> **Questions:**
>
>
>
> > *Could the authors elaborate on why TreeGRPO performs less favorably on certain reward metrics such as ImageReward and CLIPScore? Does this limitation stem from the local advantage propagation design or other factors like the tree structure or sampling schedule?*
>
> TreeGRPO’s reduced gains on metrics like ImageReward and CLIPScore primarily reflect reward characteristics and training design, not a limitation of the tree structure or local advantage propagation. Importantly, we train on specific rewards (e.g., HPS) and evaluate on others; the model is not optimized toward metrics like ImageReward. Despite this, TreeGRPO achieves the best overall performance across metrics while improving training efficiency.
>
>
>
> > *There is a concurrent work named BranchGRPO [1] that appears closely related to TreeGRPO in terms of motivation and overall design. Could the authors briefly compare TreeGRPO with BranchGRPO?*
>
>
>
> Our proposed **TreeGRPO** and the concurrent **BranchGRPO** represent two distinct approaches to tree-based reinforcement learning, with the following key differences:
>
> - **Branching Strategy**: BranchGRPO employs fixed splitting patterns, while TreeGRPO utilizes a dynamic random window approach, offering complementary advantages in different scenarios.
> - **Leaf Advantage Calculation**: TreeGRPO adopts group normalization for leaf advantage estimation, contrasting with the depth-wise normalization in BranchGRPO.
> - **Edge Advantage Formulation**: We develop a log-probability weighted reward scheme in TreeGRPO, whereas BranchGRPO uses a fused reward approach.
> - **Pruning Strategy**: TreeGRPO eliminates pruning entirely, providing a simplified alternative to the width+depth pruning mechanism in BranchGRPO.
>
> | **Aspect**         | **BranchGRPO**      | **TreeGRPO**      |
> | ------------------ | ------------------- | ----------------- |
> | **Branching**      | Fixed splits        | Random window     |
> | **Leaf Advantage** | Depth-wise norm     | Group norm        |
> | **Edge Advantage** | Fused reward        | Log-prob weighted |
> | **Pruning**        | width+depth pruning | No-pruning        |
>
> These architectural differences highlight the diverse design choices in concurrent tree-based RL research, with each method offering unique strengths for specific applications and problem domains.
>
>
>
> Thank you again for your detailed review. We have revised our paper and updated it on the website. And we highlight the important changes and essential details that reviewers have mentioned in blue color.

---

### Official Review · Reviewer_orTx · 2025-11-01

**Soundness:** 2
**Presentation:** 3
**Contribution:** 3
**Rating:** 4
**Confidence:** 3

**Summary:**

This paper introduces an approach to fine-tuning diffusion models through a search tree.

**Strengths:**

The paper tackles the problem of making RL-based fine-tuning of vision-based generative models more efficient, which is a well-motivated and common problem in the existing literature. Additionally, TreeGRPO presents a significant improvement in runtime against the baselines considered, while matching or improving performance.

**Weaknesses:**

My main concern is the following, with some additional questions/concerns listed below. In Section 3.3., the paper claims that “Deterministic ODE solvers lack the transition probabilities required by policy-gradient RL...” and “we convert the probability-flow ODE... to an equivalent SDE that admits tractable likelihoods...”. Likelihoods necessary for RL are not tractable in SDE’s (e.g. due to the Brownian motion)? How is the paper computing log likelihoods for the advantage-based update in Equation 13?

Additionally, the paper claims an improvement in sample efficiency, but the results are in terms of runtime. In Figure 1, how does the training on 8 GPUs correspond to the runtime on a single GPU? Are these different training setups? There is limited explanation of the implementation details provided. No appendix was provided in the PDF.

**Questions:**

See Weakness section.

---

> ### Author Response · Authors · 2025-11-20
> **Response to Reviewer orTx**
>
> We sincerely thank you for your review and your recognition of its strengths, including our novel approach, comprehensive evaluation, and clear presentation. Below, we provide detailed responses to your comments.
>
>
>
> **Weakness:**
>
> > *My main concern is the following, with some additional questions/concerns listed below. In Section 3.3., the paper claims that “Deterministic ODE solvers lack the transition probabilities required by policy-gradient RL...” and “we convert the probability-flow ODE... to an equivalent SDE that admits tractable likelihoods...”. Likelihoods necessary for RL are not tractable in SDE’s (e.g. due to the Brownian motion)? How is the paper computing log likelihoods for the advantage-based update in Equation 13?*
>
> Our use of the SDE follows the same practice as previous works such as DDPO/DanceGRPO: the SDE adds a Gaussian noise term to the model-predicted mean at each sampling step. This makes every step stochastic, and the RL likelihood is computed only with respect to that sampled Gaussian noise (which is sampled from a Gaussian distribution).
>
>
>
> >  *Additionally, the paper claims an improvement in sample efficiency, but the results are in terms of runtime. In Figure 1, how does the training on 8 GPUs correspond to the runtime on a single GPU? Are these different training setups? There is limited explanation of the implementation details provided. No appendix was provided in the PDF.*
>
> Thank you for pointing out the ambiguity. Our claim of improved sample efficiency refers to the fact that, for a fixed number of model forward passes (i.e., environment steps in RL terms), TreeGRPO achieves higher reward than GRPO baselines. Wall-clock runtime is used only as a proxy for this, since all methods were run under identical throughput settings. To clarify the reviewer’s question: 1) All methods, including those in Figure 1, were trained on 8×A100 GPUs using the same batch size per GPU and the same gradient-accumulation settings. 2) The “single-GPU runtime” shown in Figure 1 is normalized wall-clock time: total training time divided by 8 to compare efficiency independent of hardware scale. We also modified the manuscript to make this more clear.
>
>
>
> Thank you again for your detailed review. We have revised our paper and updated it on the website. And we highlight the important changes and essential details that reviewers have mentioned in blue color.

---

### Official Review · Reviewer_tWQc · 2025-11-01

**Soundness:** 2
**Presentation:** 3
**Contribution:** 3
**Rating:** 4
**Confidence:** 3

**Summary:**

The paper proposes TreeGRPO, a tree-structured RL framework that reformulates diffusion-model denoising as a search tree, where shared prefixes and selective branching improve sample efficiency and step-wise credit assignment. The algorithm constructs trees by alternating deterministic ODE steps and stochastic SDE windows, then backpropagates rewards through branches to compute per-edge advantages for PPO style updates. Experiments on SD-3.5-Medium show TreeGRPO achieves comparable or higher alignment rewards than GRPO based baselines while cutting training time.

**Strengths:**

1.	Recasting diffusion denoising as a search tree with shared prefixes is a creative idea that directly addresses sample efficiency and credit assignment issues. The use of log probability weighted backup for per-edge advantages is theoretically sound.
2.	In terms of efficiency gains, TreeGRPO reduces per‑iteration training time by ~$2\times$ - $3\times$ while matching or surpassing baseline alignment scores. The method shows especially strong improvements in aesthetic scores.
3.	The paper provides a thorough empirical evaluation by comparing TreeGRPO against strong baselines across multiple reward models and both single reward and multi reward settings. Ablation studies systematically vary tree width, depth and sampling strategies, illustrating tradeoffs.

**Weaknesses:**

1.	While the method amortizes computation, branching multiple trajectories simultaneously increases memory usage, especially for large diffusion models. The paper does not quantify the computational overhead relative to baselines or provide strategies for memory management beyond acknowledging the issue.
2.	The performance improvement is somewhat marginal. Although TreeGRPO outperforms baselines on HPS and aesthetics, DanceGRPO achieves the highest ImageReward score in the single reward setting. The difference in ClipScore between TreeGRPO and baselines is small. Additional analysis could clarify why some metrics benefit less. Also, the Aesthetic metric is much easier to improve compared to others.
3.	Branching creates many edges from shared prefixes. That amortizes compute but correlates samples. So the effective on‑policy batch may be narrower than it looks. Importance weighting corrects policy drift but not correlation. Tracking effective sample size or using prefix‑level decorrelation, for example, multiple independent seeds per batch, would clarify the true statistical efficiency.
4.	The framework introduces extra hyperparameters, branching factor, tree depth, SDE window ratio, and number of trees, which significantly affect performance. Adaptive schedules are mentioned as future work but not explored.

**Questions:**

1. The tree structure seems designed for training; does it also accelerate inference or sample diversity at deployment? If not, could the branching be leveraged for generating diverse images at inference time?

2. The method combines tree search with GRPO; how much of the observed gain comes from tree exploration versus the per-edge GRPO update? Would a tree based extension of standard PPO yield similar benefits?

---

> ### Author Response · Authors · 2025-11-20
> **Response to Reviewer tWQc**
>
> We sincerely thank you for your review and your recognition of its strengths, including our novel approach, comprehensive evaluation, and clear presentation. Below, we provide detailed responses to your comments.
>
>
>
> **Weakness:**
>
> >  *While the method amortizes computation... The paper does not quantify the computational overhead relative to baselines or provide strategies for memory management beyond acknowledging the issue.*
>
> We would like to clarify that TreeGRPO does not increase peak GPU memory usage relative to GRPO baselines. Peak memory for diffusion training is dominated by (1) the batch size, and (2) whether gradients are enabled. Our tree rollouts reuse prefixes and only increase the number of forward passes, not the size of simultaneously-held activations. In practice, we cap the effective batch size to match the baseline (e.g., fixed number of latent samples per iteration) regardless of tree width. As a result, both during sampling (with gradients) and during training (no gradients), the peak activation footprint remains unchanged.
>
>
>
> >  *The performance improvement is somewhat marginal.... Additional analysis could clarify why some metrics benefit less. Also, the Aesthetic metric is much easier to improve compared to others.*
>
> We acknowledge that different reward models exhibit distinct behaviors and biases, which can sometimes conflict. For example, the original diffusion model outperforms all baselines and TreeGRPO on CLIPScore, while DanceGRPO achieves the highest ImageReward in some settings. These differences reflect that some metrics have weaker or noisier gradients and are less sensitive to step-wise credit assignment. Conversely, TreeGRPO provides the largest gains on HPS-v2.1 and Aesthetic, where improved credit assignment matters most.
> Importantly, when considering all metrics together, TreeGRPO achieves the best overall performance. As shown in Tables 1–2, it consistently ranks highest on HPS-v2.1 and Aesthetic, maintains strong ImageReward performance, and reduces iteration time by roughly 50% compared to baselines. This demonstrates that TreeGRPO delivers the most balanced improvements in both training efficiency and alignment quality, even if gains vary across individual reward types.
>
>
>
> > *Branching creates many edges from shared prefixes. .... The framework introduces extra hyperparameters, branching factor, tree depth, SDE window ratio, and number of trees, which significantly affect performance. Adaptive schedules are mentioned as future work but not explored.*
>
> We agree that the introduced hyperparameters are not ideal. In practice, they mainly govern a trade-off between performance and efficiency, as shown in Table 3: larger tree depth or branching factor can improve alignment scores but reduce efficiency. Importantly, our experiments did not require extensive tuning to achieve strong overall results, demonstrating that TreeGRPO is robust to reasonable default settings.
>
>
>
> **Questions:**
>
> >  *The tree structure seems designed for training; does it also accelerate inference or sample diversity at deployment? If not, could the branching be leveraged for generating diverse images at inference time?*
>
> TreeGRPO is designed solely to improve training efficiency. It does not reduce inference cost, because inference remains a single forward trajectory requiring the standard N diffusion steps. The branching structure does not carry over to deployment. That said, one could apply a tree-like branching strategy during inference to increase sample diversity (e.g., sampling multiple stochastic branches from shared prefixes), but this is orthogonal to our method and not part of the proposed framework. We will clarify this distinction in the paper.
>
> > *The method combines tree search with GRPO; how much of the observed gain comes from tree exploration versus the per-edge GRPO update? Would a tree based extension of standard PPO yield similar benefits?*
>
> Thank you for the question. In our setting, GRPO is essential, and simply replacing it with PPO—even with the same tree structure—does not yield comparable performance. The key reason is that PPO requires a value function to estimate advantages, but diffusion-model RL lacks a stable, generalizable value model: the state space is extremely high-dimensional (latent images), rewards are prompt-dependent, and value targets exhibit high variance across diffusion timesteps. GRPO avoids this issue by using grouped reward normalization rather than an explicit value function, making it much better suited for diffusion policies. The tree structure addresses a separate issue—credit assignment and exploration—but relies on GRPO’s stable advantage estimator.
>
>
>
> Thank you again for your detailed review. We have revised our paper and updated it on the website. And we highlight the important changes and essential details that reviewers have mentioned in blue color.

---

### Official Review · Reviewer_5wvK · 2025-11-05

**Soundness:** 3
**Presentation:** 2
**Contribution:** 2
**Rating:** 6
**Confidence:** 2

**Summary:**

The authors propose a tree-structured technique for advantage estimation and apply it to training diffusion models via GRPO.

**Strengths:**

(+) I haven't seen the idea of tree-structured advantage estimation before.

(+) I appreciate the ablations performed.

**Weaknesses:**

(-) From what I can tell, the results are not significantly stronger than similar baselines performance-wise. That said, requiring less compute is definitely a plus.

**Questions:**

1. Can you add in a comparison to https://arxiv.org/abs/2404.16767 if you have the compute?

2. Beyond just being empirically promising, I think this paper would be much stronger if there were a more compelling conceptual / theoretical reason why tree-structured advantage estimation is the "right" answer. I spent some time thinking about this but I couldn't come up with a reason why. Could you provide some thoughts on this, perhaps working things out on simple toy problems (e.g., flows between two Gaussians)?

---

> ### Author Response · Authors · 2025-11-20
> **Response to Reviewer 5wvK**
>
> We sincerely thank you for your review and your recognition of its strengths, including our novel approach, comprehensive evaluation, and clear presentation. Below, we provide detailed responses to your comments.
>
> **Weakness:**
>
> > *From what I can tell, the results are not significantly stronger than similar baselines performance-wise. That said, requiring less compute is definitely a plus.*
>
> We agree that TreeGRPO’s performance improvements are modest on certain metrics compared to strong baselines. However, a key advantage is that it reduces per-iteration training time by roughly 50%, as shown in Tables 1–2, without sacrificing alignment quality on most rewards. This efficiency gain is especially valuable for large diffusion models, making TreeGRPO a practical alternative even when absolute metric improvements are modest.
>
> **Questions:**
>
> > *Can you add in a comparison to https://arxiv.org/abs/2404.16767 if you have the compute?*
>
> Thank you for the suggestion. Our focus is on GRPO-based training to make the contribution concrete and self-contained. The method in arXiv:2404.16767 is largely orthogonal, targeting a novel optimization via regression rather than training efficiency or credit assignment. Therefore, we think a direct comparison is not necessary. However, we do think that our method can be combined with their method to boost the performance, we will note this in our paper.
>
>
>
> >  *Beyond just being empirically promising, I think this paper would be much stronger if there were a more compelling conceptual / theoretical reason why tree-structured advantage estimation is the "right" answer. I spent some time thinking about this but I couldn't come up with a reason why. Could you provide some thoughts on this, perhaps working things out on simple toy problems (e.g., flows between two Gaussians)?*
>
> We appreciate the reviewer's insightful suggestion to strengthen the theoretical grounding of TreeGRPO. Beyond its empirical success, we have formalized why the tree-structured advantage estimation is the "right" answer from two complementary perspectives: **Optimization Stability** and **Robustness Regularization**.
>
> - **Variance Reduction via Rao-Blackwellization (Optimization Perspective)**:  Standard RL methods rely on single-trajectory Monte Carlo estimates, which suffer from high variance due to the stochastic nature of diffusion models. We prove that TreeGRPO's log-probability weighted averaging acts as a Rao-Blackwellized estimator. By aggregating information from $K$ branches, the variance of the advantage estimator is reduced by a factor proportional to the effective sample size (ESS):
> $$\text{Var}(\hat{A}_{\text{tree}}) \approx \frac{\sigma^2_{\text{env}}}{\text{ESS}} \le \text{Var}(\hat{A}_{\text{single}})$$
>
>    This variance reduction tightens the lower bound on policy improvement, directly explaining the observed faster convergence.
> - **Implicit Smoothness Regularization (Robustness Perspective)**:  We demonstrate that optimizing the tree-averaged objective is mathematically equivalent to maximizing a standard objective with an additional penalty on the curvature (Hessian) of the reward landscape:
>   $$\mathbb{E}_{a \sim \pi}[Q(s, a)] \approx Q(s, \mu) + \frac{1}{2} \text{Tr}(\nabla^2 Q)$$
>    By explicitly sampling the local neighborhood, TreeGRPO penalizes "sharp peaks" (brittle solutions dependent on specific noise seeds) and encourages the policy to find "wide basins" of high reward. This justifies why our method establishes a superior Pareto frontier compared to baselines.
>   We have added a new section **Theoretical Analysis** to the main paper detailing these derivations.
>
> Thank you again for your detailed review. We have revised our paper and updated it on the website. And we highlight the important changes and essential details that reviewers have mentioned in blue color.

---

### Meta-Review · Area_Chair_KQMC · 2026-01-08

**Summary:**

The rebuttal addressed most major technical concerns and improved clarity, particularly around likelihood computation, efficiency claims, memory usage, and theoretical motivation. Remaining concerns primarily relate to the incremental nature of performance improvements and sensitivity to reward choice, rather than correctness or novelty. While performance gains are sometimes modest and not uniform across metrics, the efficiency improvements, methodological novelty, and thorough experimental validation justify acceptance.

**Reviewer Concerns:**

The rebuttal effectively addressed the need for theoretical justification by adding a new section on variance reduction via Rao-Blackwellization and implicit smoothness regularization, with derivations explaining faster convergence and robustness. It also clarified why a direct comparison to the arXiv:2404.16767 paper is orthogonal and suggested potential combinations. Concerns about marginal performance gains were partially addressed by emphasizing efficiency as a key practical advantage, especially for large models, and noting strong improvements on aesthetics. Questions on inference-time benefits and the relative contributions of tree exploration vs. per-edge updates were answered thoughtfully, highlighting training focus but potential for inference diversity, and attributing gains to both components.

Outstanding concerns include unquantified memory overhead (acknowledged but not mitigated), potential sample correlation from shared prefixes (importance weighting helps but doesn't fully decorrelate), and the introduction of extra hyperparameters without exploration of adaptive schedules, which could affect reproducibility and tuning effort.

**Reviewer Scores:**

(1) Reviewer 5wvK (initial: 6): Likely to maintain or slightly increase the score to 7 following the added theoretical justification and clarification, though still viewing gains as incremental.
(2) Reviewer tWQc (initial: 4): Likely to increase modestly to around 5 after rebuttal, as major technical and memory-related concerns were addressed, but reservations about marginal gains and correlation remain.
(3) Reviewer orTx (initial: 4): Likely to increase to approximately 5–6, given satisfactory clarification of likelihood computation and efficiency claims.
(4) Reviewer 8di4 (initial: 6): Likely to maintain the current score, as core strengths remain unchanged and concerns about metric sensitivity and hyperparameter dependence persist, albeit acknowledged.

---

### Decision · Program_Chairs · 2026-01-26

Accept (Poster)